# Effects of Sodium Arsenite on the Myocardial Differentiation in Mouse Embryonic Bodies

**DOI:** 10.3390/toxics11020142

**Published:** 2023-02-01

**Authors:** SunHwa Jeong, Changhwan Ahn, Jin-Sook Kwon, KangMin Kim, Eui-Bae Jeung

**Affiliations:** 1Laboratory of Veterinary Biochemistry and Molecular Biology, College of Veterinary Medicine, Chungbuk National University, Cheongju 28644, Republic of Korea; 2Laboratory of Veterinary Physiology, College of Veterinary Medicine, Jeju National University, Jeju 63243, Republic of Korea; 3Veterinary Medical Research Institute, Jeju National University, Jeju 63243, Republic of Korea

**Keywords:** sodium arsenite, embryotoxicity, cardiomyocyte differentiation, mitochondrial dysfunction, embryonic stem cell

## Abstract

Arsenic in inorganic form is a known human carcinogen; even low levels of arsenic can interfere with the endocrine system. In mammalian development, arsenic exposure can cause a malformation of fetuses and be lethal. This study examined the effects of sodium arsenite (SA) as the inorganic form of arsenic in embryonic bodies (EBs) with three germ layers in the developmental stage. This condition is closer to the physiological condition than a 2D cell culture. The SA treatment inhibited EBs from differentiating into cardiomyocytes. A treatment with 1 μM SA delayed the initiation of beating, presenting successful cardiomyocyte differentiation. In particular, mitochondria function analysis showed that SA downregulated the transcription level of the Complex IV gene. SA increased the fission form of mitochondrion identified by the mitochondria number and length. In addition, a treatment with D-penicillamine, an arsenic chelator, restored the beat of EBs against SA, but not mitochondrial dysfunction. These findings suggest that SA is a toxicant that induces mitochondrial damage and interferes with myocardial differentiation and embryogenesis. This study suggests that more awareness of SA exposure during pregnancy is required because even minuscule amounts have irreversible adverse effects on embryogenesis through mitochondria dysfunction.

## 1. Introduction

Arsenic pollution is found in groundwater, abandoned mines, or through agricultural activity [1,2,3]. Humans can be exposed to arsenic directly from the arsenic dissolved in water or indirectly from plants and animals exposed to arsenic polluted soil [1,4,5,6]. For example, the tobacco plant absorbs arsenic directly from the soil. Hence, first- and second-hand smoking results in direct arsenic uptake [7]. The WHO, in 2017, in the standard from the JECFA (Joint FAO/WHO Expert Committee on Food Additives) PTWI (Provisional Tolerable Weekly Intake), stated a provisional drinking water guideline for arsenic of 10 μg/L. A subsequent re-evaluation of the JECFA required lowering the standard, but the WHO did not update it because it was difficult to meet the new standard in most countries [8]. Attempts to remove arsenic from the earth by soil clearance and chemical treatment isolation have been reported, but accidents constantly occur, resulting in arsenic exposure [9].

Arsenic was reported to cause developmental neurotoxicity, even in minuscule amounts [10]. An in vitro study reported that arsenic in μM concentrations affects cell death and the molecular mechanism of the inner cellular level. Exposing immortalized mesencephalic cells to only 10 μM sodium arsenite (SA) increased the nuclear factor-κB and activator protein-1 activities [11]. In addition, low-level arsenic exposure (0.25–1 μM) caused apoptosis in pancreatic β-cells by inhibiting Thioredoxin reductase activation [12] and induced carcinogenesis in human liver cells [13]. In patients diagnosed with cardiovascular disease, nail arsenic levels above 1.38 μg/g were also associated with an increased risk of cardiovascular disease [14]. The cardiovascular effects of chronic low-dose exposure to arsenic are unknown, and the current epidemiologic studies have been unable to answer this question.

The mitochondria play an essential role in cardiac development [15]. Compared to other tissues, cardiac cells consume more energy during contraction and excitation, requiring more mitochondria [16]. The mitochondria are a vital organelle governing proliferation, differentiation, maintenance of stemness, and self-renewal [17,18]. A close relationship between the function and role of morphological changes to the mitochondria has been reported [19]. When the mitochondria are exposed to toxicants, the number increases or decreases to protect the energy-producing mechanisms [20,21,22]. Mitochondrial fusion is not necessary for cell survival, but it is essential in the developmental stage [19]. In a few studies, SA was reported as a mitochondrial toxicant in mouse primary hepatocytes [23], human bronchial epithelial cells [24], and HeLa cells [25]. Over the past few decades, SA has been reported to have developmental and reproductive toxicity in animal models, including zebrafish and rodents. Rebuzzini P. et al. reported the alternative effects of arsenic trioxide, which is an inorganic compound and medication to treat cancer [26]. Arsenic trioxide interrupted the differentiation of embryonic stem cells (ESCs) to cardiomyocytes. On the other hand, the SA effect on mitochondria function is not known in mammalian development. In vitro, primary ESCs cultured from blastocyst differentiate to embryonic body (EB) structures, including ectoderm, mesoderm, and endoderm [27]. In addition, under specific culture conditions, the EB differentiates into cardiomyocytes, including sinus nodal, atrial, and ventricular cell types [28]. Therefore, a study of ESCs and EBs is needed to determine the effects of toxicants on embryogenesis, including cardiac development.

This study examined the developmental toxicity of SA at the in vitro level using mouse ESCs and EBs. The main focus was on the effects of SA on the mechanism of embryo-to-cardiomyocyte differentiation.

## 2. Materials and Methods

### 2.1. Cell Culture and Chemical Treatment

ES-E14TG2a, a mouse embryonic stem cell line (mESCs), was purchased from the American Type Culture Collection (ATCC; Manassas, VA, USA) and cultured using the same method described elsewhere [29]. The composition of the growth medium for maintaining undifferentiated cells was as follows: DMEM/F-12 (1:1) (11320-033, Gibco, Grand Island, NY, USA), 10% FBS (16000-044, Gibco, Grand Island, NY, USA), non-essential amino acid (NEAA; 11140-050, Gibco, Grand Island, NY, USA), plasmocin prophylactic (Plas; ant-mpp, InvivoGen, San Diego, CA, USA; 50 μg/mL), 100 U/mL penicillin and 100 mg/mL streptomycin (P/S; L0022, Biowest, Riverside, MO, USA), 2-mercaptoethanol (21985-023, Gibco, Grand Island, NY, USA; 10^−4^ M), and mouse leukemia inhibitory factor (mLIF; ESG1107, Millipore, Darmstadt, Germany; 10 ng/mL). Mouse embryo fibroblasts (mEFs) were used as the feeder cells. The mEFs were obtained from E10.5 mouse embryos [30]. The cells were grown at 37 °C in a 5% CO_2_ humidified tissue culture incubator (MCO-18AIC, Sanyo, Osaka, Japan). The 3T3-L1 cells (Clone A31 and ATCC) for the general toxicity test were cultured in DMEM (LM 001-05, WELGENE, Gyeongsan-si, Gyeongsangbuk-do, South Korea) with 10% fetal bovine serum (FBS; S1480-500, Biowest, Nuaillé, France) and P/S.

### 2.2. Measurement of Cardiogenic Differentiation in the Embryonic Body

Differentiation of mESCs to the myocardium was conducted, as described elsewhere [31]. Briefly, to form the embryonic bodies (EBs), one drop had a volume of 20 μL and contained 800 cells. Eighty-four drops were suspended upside down in a petri dish (10090, SPL, Pocheon-si, Gyeonggi-do, South Korea) and filled with D-PBS at the downside to prevent evaporation. The cells were cultured in a 5% CO_2_ incubator at 37 °C for four days. After harvesting the EBs, they were placed in 60mm petri dish for stabilization for two days. Subsequently, the EB was transferred for attachment in a 24-well plate with 0.5 mL of a differentiation medium, a component at 15% FBS concentration with no LIF in the growth medium. Sodium arsenite (SA; S7400, Sigma, Burlington, MA, USA) and D-penicillamine (DPA; P4875 Sigma, Burlington, MA, USA) were treated at this time, and the medium was changed every two days for 15 days.

The beating cardiomyocytes were observed using a phase-contrast microscope every two days. The beating ratio was regarded as the ratio of beating wells and non-beating wells; the well with beating cardiomyocytes was one, and the well with non-beating cardiomyocytes was zero. The number of independent experiments was performed with eight wells in three independent experiments.

### 2.3. Analysis of Transcriptional Expression

A Trizol reagent (AM9738, Invitrogen, Waltham, MA, USA) was used for total RNA purification. Reverse transcription was performed for cDNA synthesis using 1μg RNA, and the final volume was made up to 20 μL using an M-MLV kit (28022-013, Invitrogen, Waltham, MA, USA) and a recombinant RNase inhibitor (RRI; 2313A, TaKaRa, Kusatsu, Shiga, Japan) mixture. Gene expression was confirmed by a quantitative polymerase chain reaction (qPCR; Quantstudio 3, applied biosystems, Waltham, MA, USA) using a Prime Q-Mastermix and ROX Dye (Q-9210, GENEBIO, Busan, South Korea). qPCR was conducted under the following steps: 40 cycles of denaturation at 95 °C for 30 s, annealing at 58 °C for 30 s, and elongation at 72 °C for 30 s. The fluorescence intensity was measured at the extension phase of every single cycle. QuantStudioTM Design & analysis Software ver.1.4.1 was used for template design and ΔCt value analysis. The expression levels of the genes were compared quantitatively using the ∆∆Ct method with 18S as the endogenous control for expression. Details of the experimental method can be found elsewhere [32]. For statistical confidence, the Ct values were obtained from the triplicate average. Appendix A lists the primers used.

### 2.4. Measurement of Mitochondrial Function

#### 2.4.1. Measurement of Mitochondrial ROS (mtROS)

The occurrence of mtROS was measured using mitoSOX dye (M36008, Invitrogen, Waltham, MA, USA; 5 μM) and Hoechst 33342 (H3570, Invitrogen, Waltham, MA, USA; 12 μg/mL) for 10 min at 37 °C. The cells were washed three times with D-PBS to determine the clear fluorescence absorption intensity using a SYNERGY H1 microplate reader (Agilent Bio Tek, Winooski, VT, USA).

#### 2.4.2. Measurement of Mitochondrial Membrane Potential

The mitochondrial membrane potential (ΔΨ) was measured. The drug treatment conditions were the same as those for measuring the cytotoxicity. The cells were washed with D-PBS and then exposed to Jc-1 dye (T3168, Invitrogen, Waltham, MA, USA; 10 mg/mL) for 15 min at 37 °C. The cells were washed three times with D-PBS to determine the fluorescence absorption intensity using a SYNERGY H1 microplate reader.

#### 2.4.3. Imaging of Live Mitochondria

Before the attachment of EBs, four-well-chamber slides (154526, Nunc, Rochester, NY, USA) were coated using poly-D-lysine (P647-5MG, Sigma, Burlington, MA, USA; 100 μg/mL) and laminin (354232, Corning, Corning, NY, USA; 0.4 μg/mL) for 1 h in a 37 °C incubator. For mitochondria staining, three EBs per well were cultured in an SA-containing medium for five days. The mitochondria were stained by MitoTracker Red CMXRos (9082, Cell Signaling Technology, Danvers, MA, USA; 100 nM) for 30 min at 37 °C after washing with D-PBS and followed by the direction for fixation using ice-cold methanol for 15 min at −20 °C. Subsequently, the cells were stained with 4′,6-diamidino-2-phenylindole (DAPI; D1306, Invitrogen, Waltham, MA, USA; 30:1 dilution) and protected from light for 10 min at RT. The cells were also washed three times with D-PBS. Finally, the cells were mounted in Fluoro-Gel (17985-11, Electron Microscopy Sciences, Hatfield, PA, USA) and captured with a phase-contrast microscope (BX51, Olympus, Shinjuku, Tokyo, Japan) with 400× magnification. Five pictures, at least ten cells per picture, were analyzed using multi-point tools for mitochondria counting and a freehand line for mitochondrial fusion or fission in cells in Image J software (National Institutes of Health, Bethesda, MD, USA).

### 2.5. Evaluation of Embryotoxicity Using EB Test Method

The cells were seeded on 96-well plates with 800 cells/well in 50 μL of culture media. After incubation for two hours, SA was diluted 1.333-fold and used to produce a drug badge and added at 150 μL/well. After culturing in an incubator for four days, a CCK assay was performed using an EZ-cytox solution (EZ-500, DoGenBio, Seoul, South Korea). All existing drug badges were removed, and the cells were washed once with D-PBS. Subsequently, 110 μL of a mixture of pure DMEM/F12 (1:1) media and EZ-cytox solution at a 10:1 ratio was added. The resulting mixture was incubated for one hour. After incubation, the absorbance at 450 nm was measured using a SYNERGY H1 microplate reader (Biotek instrument, Winooski, VT, USA). The cell viability was assessed by comparing the relative absorbances with that of the solvent control set to 100%. The IC_ic_, which means 50% cell viability, was substituted for the value of the y-axis in the trend line formula of the graph.

The EBs prepared, described as written above, were incubated in eight concentrations of SA and in PBS as a vehicle for four days. The image of EBs was captured using a phase-contrast microscope (IX71, Olympus, Shinjuku, Tokyo, Japan) with 100× magnification and phase 1. The image contained at least five EBs. Eight images were gathered, and the radius of the EB area was measured using Image J software.

### 2.6. Statistics

The significant differences were detected using ANOVA followed by a Tukey’s test for multiple comparisons. The analysis was performed using the Prism Graph Pad v8.0.1 (Graph Pad Software, San Diego, CA, USA). The values are expressed as the mean ± standard deviation (SD) of at least three separate experiments; *p* values < 0.05 were considered significant.

## 3. Results

### 3.1. Sodium Arsenite Disrupted a Myocardial Differentiation of Embryonic Body

Myocardial differentiation was induced by replating the EBs from a petri dish to a 24-well plate, as shown in the experimental scheme (Figure 1A). The following experiments were performed at SA concentrations of 10^−8^–10^−6^ M. This concentration range of inorganic arsenic is considered to pollute [33]. Two days after replating, the EB of the control group (Figure 1B) was attached to the plate, whereas some of the EB did not attach to the plate after SA exposure (Figure 1C). In this study, two-day incubation was sufficient to attach EB in the control group. Exposure to 10^−6^ M SA decreased the attachment efficacy significantly compared to the control group (Figure 1D), and 10^−7^ and 10^−8^ M SA had no effect (data not shown). The effect of SA on myocardial differentiation was evaluated by the appearance of beating cardiomyocytes. Beating is a prominent feature of cardiomyocytes. From the second day, the beating ratio showed a marked decrease at 10^−6^ M SA on induced differentiation after ten days (four days after attachment, Figure 1E).

The transcriptional levels of cardiac differentiation-related genes were measured on Days 4, 10, and 15 after SA (10^−6^ M) exposure. The SA treatment significantly decreased the expression of the brachyury gene, a mesodermal marker, on Day 4 but not on Days 10 and 15 (Figure 2A). In addition, the expression levels of the cardiac sodium-calcium exchanger (*Ncx1*) gene and cardiac troponin I (*cTn1*) gene, which are myocardium locomotion regulators, were decreased significantly (Figure 2B,C) by the SA treatment for all 15 days. On the other hand, there were no significant changes in two transcription factors, *Gata4* and T-Box Transcription Factor 20 (*Tbx20*) (Figure 2D,E). *Tbx20* and *Gata4* are transcription factors essential for proper heart development in a growing embryo.

### 3.2. Effect of Sodium Arsenite on Mitochondria

An analysis of the mitochondrial function was conducted. Cardiomyocytes are high-energy-demanding cells. Hence, the mitochondria play an essential role in keeping the cardiomyocyte beating. First, the mitochondrial reactive oxygen species (mtROS) and mitochondrial membrane potential were measured. As shown in Figure 3A, SA 10^−7^ M and 10^−6^ M significantly increased the mtROS levels in EBs after treatment for four days. On the other hand, another marker of mitochondrial dysfunction, the mitochondrial membrane potential (ΔΨ), did not change (Figure 3B).

Mitochondria use the oxidative phosphorylation system (OXPHOS) to produce adenosine triphosphate (ATP) molecules [34,35]. Hence, dysregulation of mitochondrial OXPHOS indicates mitochondrial dysfunction. OXPHOS consists of five protein complexes: Complex I, Complex II, Complex III, Complex IV, and Complex V. The changes of mRNA expression of five complexes due to SA treatment were measured by real-time PCR at the resting EB state after SA treatment for four days. The mitochondrial mRNA expression level of NADH:Ubiquinone Oxidoreductase Core Subunit S1 (*Ndufs1*) was measured to assess the level of Complex I, Ubiquinol-Cytochrome C Reductase, Rieske iron-sulfur polypeptide 1 (*Uqcrfs1*) for Complex III, Cytochrome c oxidase subunit 5A (*Cox5a*) for Complex IV, and ATP synthase F1 subunit delta (Atp5d) for Complex V. The Complex II, which are encoded nuclear mRNA, was assessed by measuring nuclear mRNA expression of succinate dehydrogenase complex flavoprotein subunit A (*Sdha*). Interestingly, the Complex IV gene was increased significantly by the SA treatment (Figure 3F); the other four complexes were not (Figure 3C–E,G).

### 3.3. Effect of D-Penicillamine against Sodium Arsenite during Cardiac Differentiation

D-Penicillamine (DPA) is a chelating agent commonly used as an antidote in cases of accidental arsenic poisoning. This study examined whether DPA protects the EB against SA exposure. The 10^−4^ M DPA treatment defended the beating ratio decreased by the 10^−6^ M SA treatment (Figure 4A). Interestingly, the EB size, another marker of developmental toxicity, was not recovered (Figure 4B). This result cannot be fully explained by the time point difference between the beating ratio measurement and EB size measurement because the EB size was measured before the beating ratio. Therefore, this study examined whether there were any morphological changes in the mitochondria. Normal mitochondria have an appropriate length, but under stressful conditions, an abnormal morphology of a fission and fusion form has been detected. The appearance of the excessive fission and fusion form is evidence of a problem with mitochondria generation and maturation. The live mitochondria were stained red to confirm the change in mitochondrial morphology (Figure 4C). SA was treated during myocardial differentiation for 10 days. As a result, SA increased the number of mitochondria with a concomitant decrease in mitochondrial length compared to the control group, resulting in excessive mitochondrial fragmentation as the fission form. In the case of the DPA and SA co-treatment, DPA decreased the number of mitochondria and increased the length of the mitochondria compared to treatment with either in the SA-treated and control groups (Figure 4C–E). Overall, the SA treatment damaged the mitochondrial function during myocardial differentiation, and DPA had an off-target effect.

### 3.4. Change in the Gene Transcription Level by Either the SA Treatment or DPA and SA Co-Treatment during the Early EB Development Stage

The effect of a SA and DPA co-treatment with SA at the gene level was examined by comparing the gene expression of 40 genes (Figure 5A) related to pluripotency, cardiac function, and cell homeostasis (Figure 5A). As shown in Figure 5A, in the heat map using the logarithmic scale of the relative expression levels of genes, there was no noticeable single pathway change between the three groups. On the other hand, 13 genes showed significant changes (Figure 5B–N). These genes helped preserve the pluripotency and regulated the differentiation capacity of stem cells. Most genes showed marked reductions in the SA treatment, except that SA upregulated Nestin. The cell survival and homeostasis-related genes, such as *Parkin* (Figure 5J), *Chop* (Figure 5M), and *Sod2* (Figure 5N), were decreased significantly by the SA treatment. Moreover, despite the toxic effects of SA on the embryo developmental phase, the expression of the pro-apoptotic genes (*Bax*, and *Caspase 3*) and anti-apototic gene (*Bcl-2*) did not change. None of the changes by the SA treatment were protected by DPA except for the *c-Myc* and *Mef2* gene (Figure 5B,E). Furthermore, the decreasing pattern of *Pink* and Atf4 genes in the SA treatment group was strengthened by the DPA co-treatment. (Figure 5K,L).

### 3.5. Sodium Arsenite Induced Embryotoxicity

Finally, an embryonic body test (EBT) was conducted to confirm if the SA treatment showed developmental toxicity. The EBT test was developed in previous studies [29]. The test method was manufactured by three values for two endpoints: two values for the 3T3-L1 and of mESCs cell viability after the SA treatment and one for the EB formation grade. As shown in Appendix A, log IC50 is the value of the half-maximal cell survival inhibition, and the ID50 is the half inhibition concentration for EB formation during differentiation. The experimentally confirmed values were placed into a previously established discriminant function,

DF = −0.1139733 × (log10IC503T3-L1;0.511) − 0.2303571 × (log10IC50mES-Cs;0.187) − 0.6726328 × (log10ID50EB;0.171) + 2.723601 [29].

A score of 2.59 was obtained (Appendix A). The value was higher than the SDF; 2.593, confirming that SA is a developmental toxicant. Furthermore, this score is similar to a known strong developmental toxicant reported previously [29].

## 4. Discussion

Arsenic was measured at effective concentrations ≤ 0.23 ppm (~2.33 mg/kg/day) [33] in the developmental stage, and the reported NOAEL value was 2.5 mg/kg/day [36]. Across the globe, many countries drink arsenic-contaminated water ranging from 0.5 µg/L to 5 mg/L [37]. When exposed to arsenic, the detected concentration tends to increase depending on the pregnant woman’s age, and 0.701 µg/L of arsenic has been detected in the blood depending on the region. On the other hand, 0.913 µg/L of arsenic was measured in the blood of pregnant women [38]. Moreover, 0.599 µg/L of arsenic was detected in 50.9% of umbilical cord blood and 0.02 µg/L in 93.9% of the meconium in newborn babies [38].

This is the first study showing that SA adversely affects the entire differentiation period of cardiomyocytes. Several studies on humans and animals have reported arsenic as a toxicant [4,23,24,39,40,41]. On the other hand, the adverse effects of SA at early development stages are not entirely understood. In fetal development, arsenic has been studied under accidental exposure at high doses in animals, which are lethal, and the fetus is usually miscarried [42]. Only a few studies revealed the toxicity in the fetus and offspring [39,43,44]. In particular, disruptions of myocardial differentiation cause fetal death. Hence, there is technically no time to study the basal mechanism. Little is known about the effects of arsenic exposure at the embryo stage despite several types of ESCs being used to study the molecular mechanism for the effect of SA [39]. The embryo stage is built continuously by crosstalk among three germ layers: ectoderm, endoderm, and mesoderm. Thus, a 2D cell culture system using a single cell line cannot truly assess the physiological and pathological alterations. One study used zebrafish eggs for embryogenesis, but these are not mammals [45]. This study showed the adverse effects of SA using an embryonic body with three germ layers in the developmental stage without animal sacrifice. This condition is closer to the physiological condition than a 2D cell culture. Even 1 µM SA inhibited the growth of EBs, increased the mtROS, and disrupted myocardial differentiation. The present results showed that 1 µM of SA had a toxic effect on development and differentiation support in that the concentration was lower than the NOAEL value and a previous report suggested an effective low dose of 1 µM [11,12,13,23,41].

Mitochondrial DNA (mtDNA) damage and oxidative stress occur when mammalian somatic cells are exposed to SA. The mitochondrial mass increases in the cardiac developmental stage [15,46]. Mechanistically, an increase in the mtDNA copy number imparts a survival advantage to cancer cells. Furthermore, the increase in mitochondrial content elevates mitochondrial oxidative phosphorylation [47]. During differentiation, *Sod2* expression increases to remove superoxide and protein oxidation [15,48], but exposure to SA reduces *Sod2* expression. *Sod2* reduction results in increased mitochondrial oxidative stress. The level of ROS was increased by SA treatment (Figure 3A). Furthermore, this decrease in mitochondrial complex IV gene expression (Figure 3F) and increase in the mitochondrial fission form (Figure 4D,E) may be considered a mitochondrial dysfunction. Overall, these results showed that SA exposure during embryogenesis exhibits dose-dependent mitochondrial toxicity in both short- and long-term exposure.

Another key finding is that SA exposure during cardiomyocyte differentiation disrupts heart muscle locomotion. In Figure 2A, the brachyury gene, which is expressed at the mesoderm state, tends to increase and then decrease gradually. The present results support previous reports that the *brachyury* gene is expressed mostly during the early stages of differentiation [26,49,50]. Because three germ layer markers including *nestin* and *Sox17* are expressed differentially in the early stage, it may suggest that interrupted cardiomyocyte differentiation. In this study, SA inhibits Ncx1 and cTn1 expression. During embryogenesis, *cTn1* expression in cardiac differentiation showed a different result from another arsenic compound (As2O3), suggesting that the inhibition effects are different when exposed to other arsenic structures [26,51].

D-penicillamine (DPA) is a chelator to be used in the treatment of arsenic poisoning. On the other hand, the mechanism was not established and rarely reported in clinical treatment [52]. A recent study showed Wilson’s disease-related treatment [53,54] and lead poisoning recovery [55]. Interestingly, this study showed that DPA preserved the beating of EBs (Figure 4A) but failed to restore the impairment at the morphology (Figure 4B) and gene level (Figure 5). Furthermore, in the mitochondrial form, there was no recovery of abnormal mitochondria, but DPA increased the fusion form, which was judged by a decrease in the number of mitochondria and an increase in length. There are also some results worth noting. The change in *Klf4* (pluripotency), *Mef2* (cardiac function), *Sox17* (endodermal), and *Nestin* (ectodermal) by the SA treatment were attenuated when co-treated with SA and DPA. This suggests that DPA acts as a suppressor of other differentiations. Moreover, expression of the *Ryr2* (CA2+ and gap junction), *Pink* (mitophagy), and *Atf4* (ER stress) genes were decreased more than in those treated unilaterally with SA. Hence, DPA may directly affect SA and influence intercellular organelles, such as the ER. In addition, DPA alone is also a developmental toxicant [30]. Therefore, more study on DPA is needed.

Briefly, this study examined the adverse effects of SA exposure on the cardiomyocyte differentiation of ESCs. In conclusion, even minuscule concentrations of SA can cause irreversible dysplasia or malformations during embryonic development. Hence, there is a need to reconsider the safe dose for maternal exposure.

## Figures and Tables

**Figure 1 toxics-11-00142-f001:**
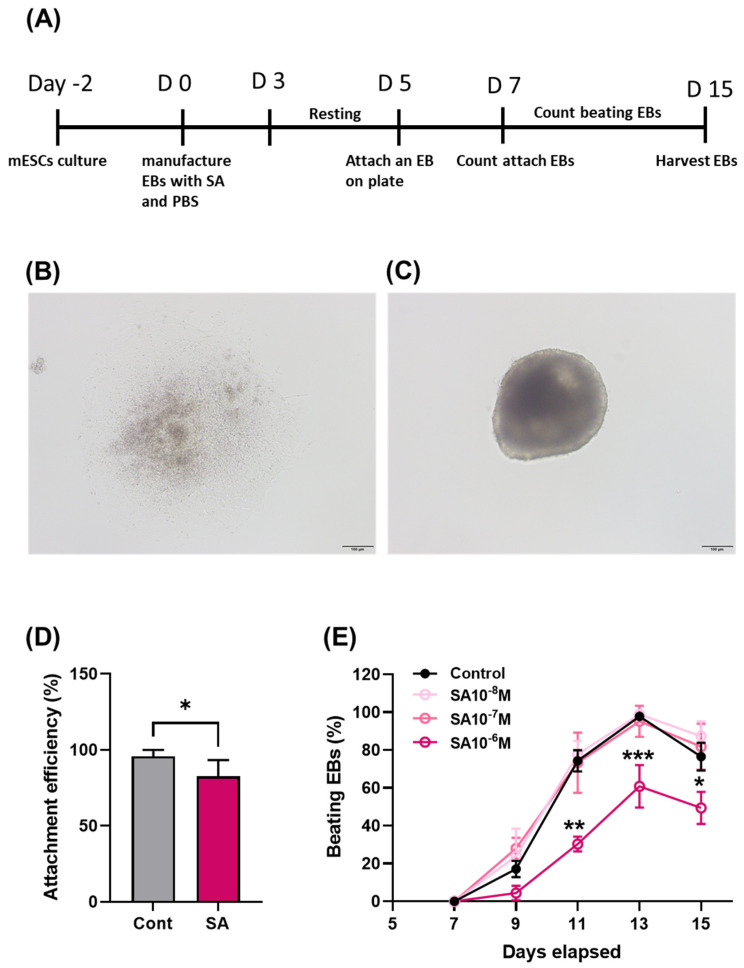
Disruption of myocardial differentiation by sodium arsenite. The cells were treated with SA during cardiomyocyte differentiation. (**A**) Attached form and (**B**,**C**) unattached form of EBs two days later. Scale bar = 100 μm. (**D**) Attachment efficiency when inducing cardiomyocyte differentiation two days later (n = 24). (**E**) Time- and dose-dependent cardiomyocyte beating ratio (n = 24). The significance was determined by ANOVA. * *p* < 0.05, ** *p* < 0.01, and *** *p* < 0.001 vs. control. * Control vs. SA 10^−6^ M, # SA 10^−7^ M vs. SA 10^−6^ M, and & SA 10^−8^ M vs. SA 10^−6^ M. Each value is expressed as means ± SD. n = 3, Cont; control, SA; sodium arsenite.

**Figure 2 toxics-11-00142-f002:**
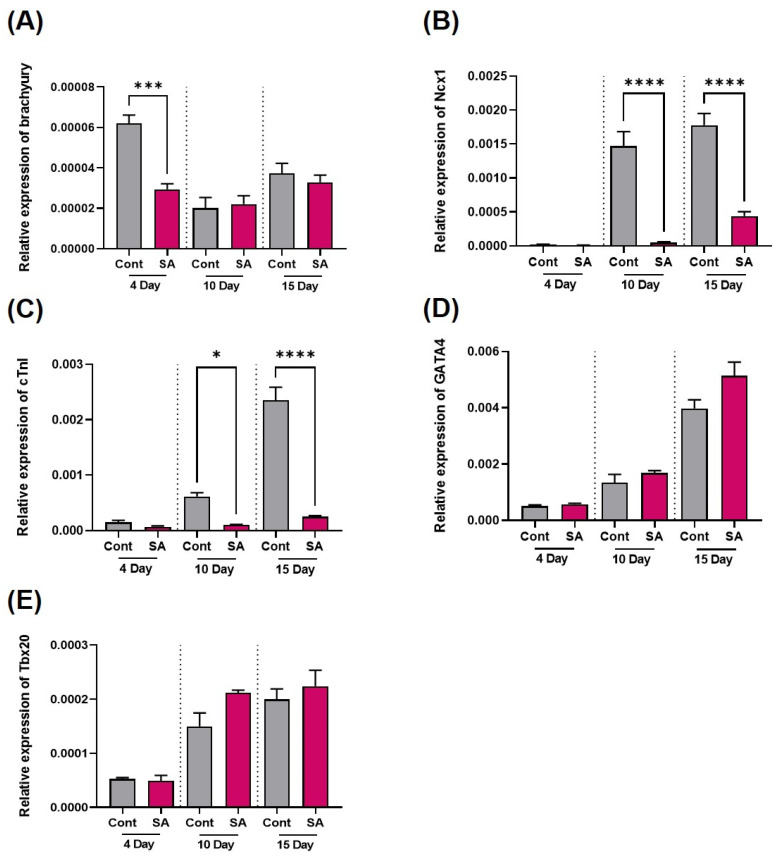
Effects of sodium arsenite on myocardial differentiation-related gene expression. Cardiac marker gene expression was measured on the 4th, 10th, and 15th days after SA treatment by real-time PCR in differentiation-induced EB to cardiomyocytes. (**A**–**E**) Bar graphs of gene transcription level, (**A**) Brachyury gene (*Bra*); (**B**) *Ncx1* gene; (**C**) Cardiac troponin I (cTnI) gene; (**D**) *Gata4* gene, and (**E**) *Tbx20* gene. The significance was obtained by ANOVA. * *p* < 0.05 and *** *p* < 0.001, **** *p* < 0.0001 vs. control. Each value is expressed as means ± SD. n = 3 in three independent experiments. Cont; control, SA; sodium arsenite.

**Figure 3 toxics-11-00142-f003:**
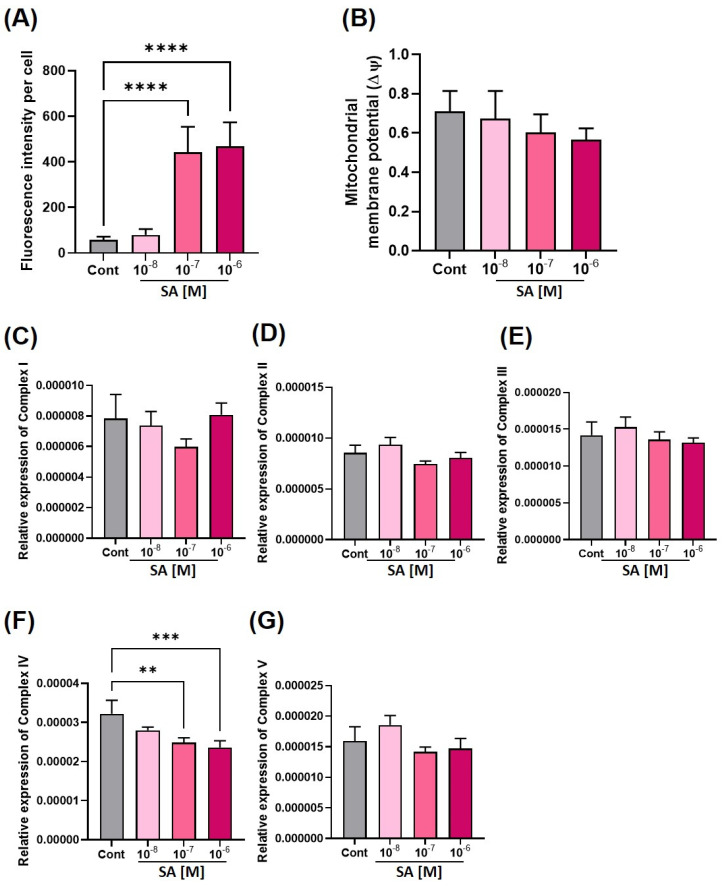
Change in the mitochondrial function and mitochondria-related gene expression in EB. The mitochondrial dysfunction during SA exposure in the EB state for four days was measured. (**A**) Bar graph of mitochondrial superoxide using mitoSOX staining, and (**B**) Bar graph of the mitochondrial membrane potential by Jc-1 staining. (**C**–**G**) Bar graphs show the gene expression levels of (**C**) Complex I, (**D**) Complex II, (**E**) Complex III, (**F**) Complex IV, (**G**) Complex V. The significance was obtained by ANOVA test. ** *p* < 0.01, *** *p* < 0.005 and **** *p* < 0.0001 vs. control. Each value is expressed as the means ± SD. n = 6 (**A**,**B**) n = 3 (**C**–**G**) in three independent experiments. Cont; control, SA; sodium arsenite.

**Figure 4 toxics-11-00142-f004:**
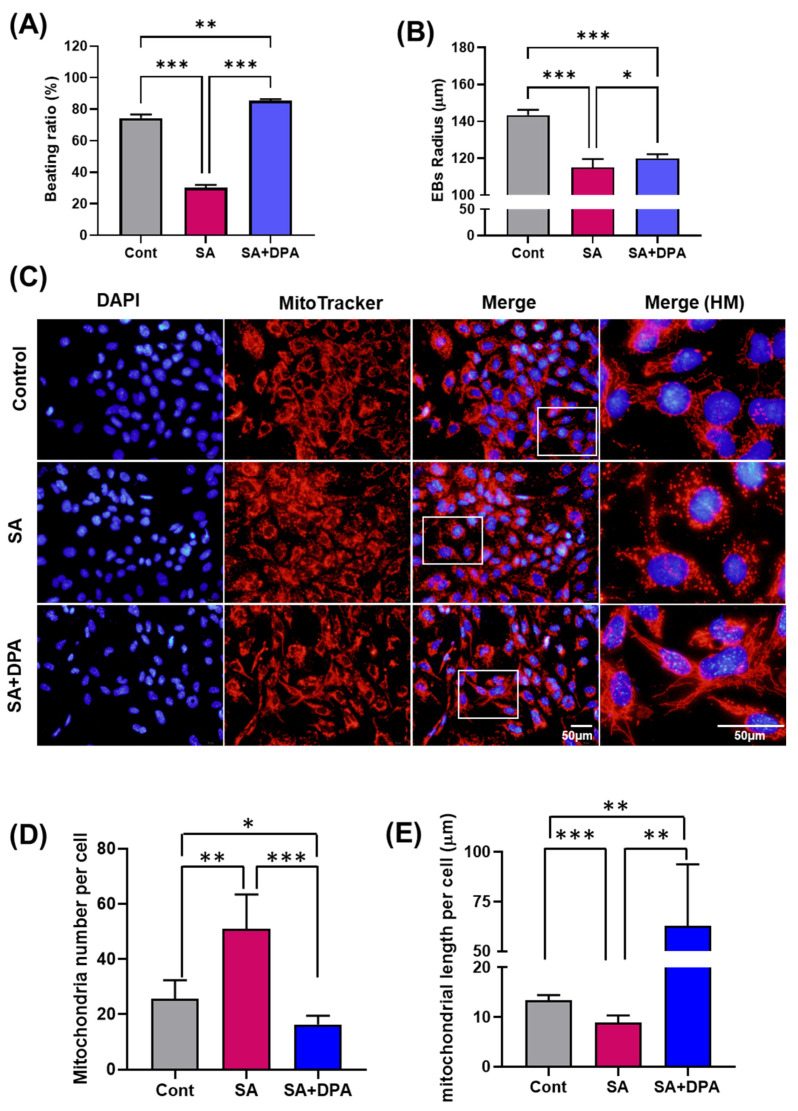
Effect of D-penicillamine against sodium arsenite during cardiac differentiation. The effect of D-penicillamine (DPA) against the interruption of cardiac differentiation by sodium arsenite (SA) exposure was measured, and the mitochondria morphology was analyzed in EB. (**A**,**B**) are bar graphs of beating ratio and EB size. (**C**) Representative pictures of mitochondria. Mitochondria (39) were stained with MitoTracker, and the nuclei (blue) were stained with DAPI. (**D**,**E**) are the bar graphs of the number of mitochondria in each cell and the mitochondria length (n > 50 cells). The significance was obtained using a Tukey’s test. * *p* < 0.01, ** *p* < 0.01, *** *p* < 0.0001 vs. control. Each value is expressed as the means ± SD. Cont; control.

**Figure 5 toxics-11-00142-f005:**
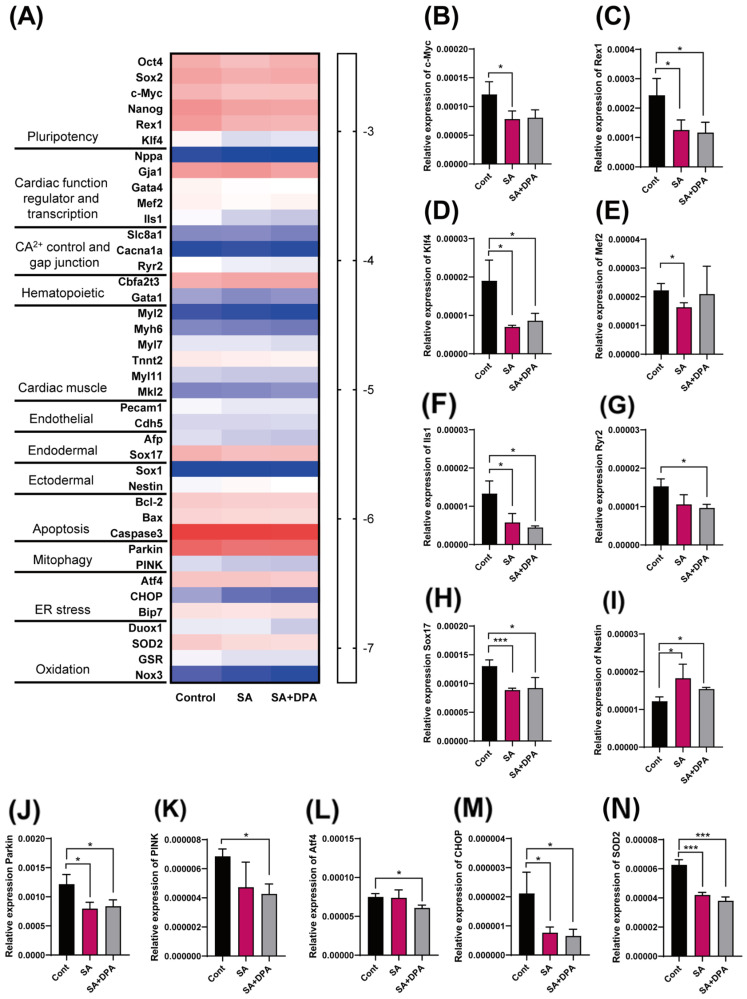
Changes in cardiac differentiation and cell survival-related gene expression levels. The transcription levels of 40 genes were measured in EB after either the SA-only treatment or the SA/DPA co-treatment with SA for four days by real-time polymerase chain reaction (qRT-PCR). (**A**) Heat map for the logarithmic scale of expression levels of comparing 40 genes. (**B**–**N**) are bar graphs of the relative expression level of the genes, which changed significantly. Significance was obtained using a student’s *t*-test. * *p* < 0.05, *** *p* < 0.001 vs. control. Each value is expressed as the means ± SD. n = 3, Cont; control, SA; sodium arsenite, DPA; D-penicillamine.

## Data Availability

Not applicable.

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
