# Peer review of "Effects of Sodium Arsenite on the Myocardial Differentiation in Mouse Embryonic Bodies"

_toxics, 2023, doi:10.3390/toxics11020142_

Round 1

Reviewer 1 Report

In this research, the authors investigate mitochondrial dynamics. It is curiously why the authors, among a large number of genes, did not study the expression of genes that are responsible for fission/fusion, such as Opa, Drp, Mfn1,2, etc.

The authors discuss the change in the number of copies of mitochondria, but did not investigate the expression of genes that are responsible for mitochondrial biogenesis, such as Tfam, Nrf1, etc.

The authors do not use the correct nomenclature when writing the name of the genes.

DPA appears unexpectedly in the results. It is not mentioned either in the introduction (although it should have been indicated in the goal of research), and in the materials and methods.

Figure 2. Please use the standard notation for statistical significance. *p<0.05, **p<0.01, and ***p<0.001.

Line 233-236. The authors should clarify which particular subunit of each ETC complex they measured. And to clarify where they are encoded, in nuclear DNA or mtDNA.

Figure 3. It is incorrect to say the expression of the complex I/II/III/IV, if we are talking about only one of the subunits of the respiratory complex.

Figure 4. Why do authors say using a t-test when the methods indicate that Tukey's test was used for multiple comparison.

Line 296. Despite that the expression of the apoptosis-related genes (Bcl-2, Bax, and Caspase 3) did not change, authors should distinguish between prapoptotic and anti-apototic genes.

Line 48. Please use full name for the “TrxR”.

Line 70. Please use abbreviation only for the “embryonic stem cells (ESCs)”.

Author Response

Response to Reviewer 1:

We have made the following editorial changes in response to his/her insightful comments.

We tried to revise the manuscript following your comments.

Review Comments to the Author

Comment 1.

In this research, the authors investigate mitochondrial dynamics. It is curiously why the authors, among a large number of genes, did not study the expression of genes that are responsible for fission/fusion, such as Opa, Drp, Mfn1,2, etc.

: Thanks for the insightful comment. The heart is one of the organs in which the number and role of mitochondria per cell are important. We hypothesized that the action of SA might affect mitochondrial ROS generation and tried to confirm its contribution to the dysfunction. Therefore, the expression of mitochondrial dysfunction-related dyes and oxidation-related genes are closely related throughout several articles, including Ren and Chen et al. (2020) (https://doi.org/10.3389/fcell.2020.580070) over the past few decades. Also, since the morphological image is shown in figure 4 in this article, we thought that additional examinations for gene expression could be replaced with references.

Comment 2.

The authors discuss the change in the number of copies of mitochondria, but did not investigate the expression of genes that are responsible for mitochondrial biogenesis, such as Tfam, Nrf1, etc.

: Thanks for the insightful comment. Mitochondrial biogenesis ultimately affects the number and size of mitochondria, the production of ATP, and the generation of ROS. The genes mentioned by the reviewer are known to be key regulators of mitochondrial biogenesis. The factors are judged to confirm the increase or decrease of mtDNA. Data related to mtDNA are put on the page below

Analysis of mitochondrial gene expression and mtDNA copy number

For mtDNA expression analysis, the total DNA was extracted using G-DEXTM-IIc (17231, iNtRON, Korea). For qPCR performance, The DNA was diluted to 3 ng/μl, and 2 μl each of a DNA and DNA solution was analyzed by qPCR, as described above. The expression level of mitochondrial DNA was compared with the expression level of nuclear DNA. Nuclear DNA was selected among genes known as one copy. Primer sequence was provided in under table.

Mitochondrial DNA copy number primer

Mitochondrial DNA

F: CCCAGCTACTACCATCATTCAAGT

Rooney et al. (2015)

R: GATGGTTTGGGAGATTGGTTGATGT

Nuclear DNA

(Calbindin-D9k, control)

F: TAAAGACTATAAAAGAGCCCCTC

Sequence ID: AF293950.1

R: CTGGGGAACTCTGACTGAAT

The mtDNA copy number (mtDNA-CN), a biomarker of the mitochondrial function, was measured. Interestingly, mtDNA-CN was increased when SA exposure, but decreased with DPA.

Rooney, J. P., Ryde, I. T., Sanders, L. H., Howlett, E. H., Colton, M. D., Germ, K. E., Mayer, G. D., Greenamyre, J. T., & Meyer, J. N. (2015). PCR based determination of mitochondrial DNA copy number in multiple species. Methods in molecular biology (Clifton, N.J.), 1241, 23–38. https://doi.org/10.1007/978-1-4939-1875-1_3

Comment 3.

The authors do not use the correct nomenclature when writing the name of the genes.

: According to the reviewer’s comment, we corrected the appropriate nomenclatures.

Comment 4.

DPA appears unexpectedly in the results. It is not mentioned either in the introduction (although it should have been indicated in the goal of research), and in the materials and methods.

: According to the reviewer’s comment, we added information of DPA in the Materials and method section, and add the information for the concentration of DPA in line 256.

Comment 5.

Figure 2. Please use the standard notation for statistical significance. *p<0.05, **p<0.01, and ***p<0.001.

: According to the reviewer’s comment, we add the phase at figure 2.

Comment 6.

Line 233-236. The authors should clarify which particular subunit of each ETC complex they measured. And to clarify where they are encoded, in nuclear DNA or mtDNA.

: According to the reviewer’s comment, we add the information of where we measure the expression of ETC complex.

“OXPHOS consists of five protein complexes: Complex I (NADH dehydrogenase), Complex II (Succinate dehydrogenase), Complex III (Coenzyme Q - cytochrome c reductase), Com-plex IV (Cytochrome c oxidase), and Complex V (ATP synthase). The five complexes were measured by real-time PCR at the mitochondrial DNA of resting EB state after SA treat-ment for four days.”

Comment 6.

Figure 3. It is incorrect to say the expression of the complex I/II/III/IV, if we are talking about only one of the subunits of the respiratory complex.

:

: Thanks for the insightful comment. Up to dates, several manuscript describe the expressions of Complex I (NADH dehydrogenase), Complex II (Succinate dehydrogenase), Complex III (Coenzyme Q - cytochrome c reductase), Com-plex IV (Cytochrome c oxidase) as equal to whole complex function. The redox energy released during this process is used to transfer protons from the mitochondrial matrix to the periplasmic space that generates proton-motive force across the inner mitochondrial membrane at complex I, III, and IV (S Schüll et al., Cell death & disease, 2015, Lokendra K et al., Curr Med Chem. 2009)

Comment 7.

Figure 4. Why do authors say using a t-test when the methods indicate that Tukey's test was used for multiple comparison.

: In the manuscript, as we mentioned in the 2.6. Statistics, we used Tukey’s test for multiple comparison. Otherwise, in the pairwise comparison, we used t-test. It is mistake during writing. Thanks for the comment.

Comment 8

Line 296. Despite that the expression of the apoptosis-related genes (Bcl-2, Bax, and Caspase 3) did not change, authors should distinguish between prapoptotic and anti-apototic genes.

: According to reviewer’s comment, we distinguished pro- and anti-apoptitic genes in the context.

“Moreover, despite the toxic effects of SA on the embryo developmental phase, the expression of the pro-apoptotic genes (Bax, and Caspase 3) and anti-apototic gene (Bcl-2) did not change.”

Comment 9

Line 48. Please use full name for the “TrxR”.

: According to reviewer’s comment, we change the TrxR to full name of it.

“In addition, low-level arsenic exposure (0.25–1 μM) caused apoptosis in pancreatic β-cells by inhibiting Thioredoxin reductase activation”

Comment 10

Line 70. Please use abbreviation only for the “embryonic stem cells (ESCs)”.

: According to reviewer’s comment, we changed the embryonic stem cell to ESCs.

Reviewer 2 Report

The paper by S. Jeong et al., entitled "Effects of sodium arsenite on the myocardial differentiation in 2 mouse embryonic bodies" reports that the adverse effects of SA exposure on the cardiomyocyte differentiation of embryonic stem cells.

It requires some consideration before it can be published in a journal.  My comments are listed below.

Line 49-52: Please delete one of them since the content is duplicated.

Line 53-54: The references chosen by the author seem a bit too old; there are papers reporting the association between low arsenic exposure and cardiovascular disease since 2005 to the present. The information needs to be updated.

Line 66: A period is required between “rodents” and” Rebuzzini P et al.”

Line 66-67: Please add a reference to the following statement. “Rebuzzini P et al. reported the alternative effects of arse-66 nic trioxide, which is an inorganic compound and medication to treat cancer.”

Line237-238:Could it be that only the expression levels of the complex IV genes were significantly reduced by SA treatment?

Line254:The DPA treatment recovered the beating ratio decreased by the 10−6 M SA treatment (Fig 4 A).; Did adding DPA to cells with reduced beating due to SA exposure restore beating ratio? Or was it the result of adding SA and DPA at the same time, which was comparable to the control beating ratio? If the latter is the case, the term recovered is misleading to the reader.

Line290:In line 289 it states that the expression of 13 genes is markedly altered, and in line 290 it further states the same for C-Myc, Rex1, Klf4, Mef2, Ils1, Sox17, and Nestin. Do you singled out these genes here because I wanted to emphasize the contents of line 291? If so, it is easier to understand how to write line 293.

Line297:None of the changes by the SA treatment were restored by DPA except for the c-Myc and Mef2 gene (Fig 5 B and E).;I think it is better not to use the word restored since this description also does not mean that what was changed has been restored (since it is a simultaneous exposure).

Line366-368:In Figure 2A, the brachyury gene, which is expressed at the mesoderm state, tends to increase and then decrease gradually. The present results support previous reports that the brachyury gene is expressed mostly during the early stages of differentiation.; This explanation helped us to understand why the expression in the control group is reduced.

It seems that this is the only brachyury gene in Fig. 2 that is affected by arsenic in the initial stage (4Day). Please discuss this phenomenon further.

Line366-367:As already commented, the use of "recovered" or "restored" is not appropriate in the case of simultaneous exposure. Please revise the wording to make it less misleading, e.g., simultaneous exposure suppressed the effects of SA.

Author Response

Response to Reviewer 2:

We have made the following editorial changes in response to his/her insightful comments.

We tried to revise the manuscript following your comments.

Review Comments to the Author

The paper by S. Jeong et al., entitled "Effects of sodium arsenite on the myocardial differentiation in 2 mouse embryonic bodies" reports that the adverse effects of SA exposure on the cardiomyocyte differentiation of embryonic stem cells.

It requires some consideration before it can be published in a journal. My comments are listed below.

 Comment 1.

Line 49-52: Please delete one of them since the content is duplicated.

: According to reviewer’s comment, we deleted the duplicate sentences.

 Comment 2.

Line 53-54: The references chosen by the author seem a bit too old; there are papers reporting the association between low arsenic exposure and cardiovascular disease since 2005 to the present. The information needs to be updated.

: According to reviewer’s comment, we updated up-to-dated references.

 Comment 3.

Line 66: A period is required between “rodents” and” Rebuzzini P et al.”

: According to reviewer’s comment, the sentence has been separated accordingly.

 Comment 4.

Line 66-67: Please add a reference to the following statement. “Rebuzzini P et al. reported the alternative effects of arsenic trioxide, which is an inorganic compound and medication to treat cancer.”

: According to reviewer’s comment, We have added the reference.

 Comment 5.

Line237-238:Could it be that only the expression levels of the complex IV genes were significantly reduced by SA treatment?

: We intended that it was the only significant among the complexes. The word ‘only’ was deleted because it could be misleading.

 Comment 6.

Line254:The DPA treatment recovered the beating ratio decreased by the 10−6 M SA treatment (Fig 4 A).; Did adding DPA to cells with reduced beating due to SA exposure restore beating ratio? Or was it the result of adding SA and DPA at the same time, which was comparable to the control beating ratio? If the latter is the case, the term “recovered” is misleading to the reader.

: It was the result of adding SA and DPA at the same time. Therefore, we have replaced the term ‘recovered’ to ‘defended’ for avoid misunderstanding.

 Comment 7.

Line290:In line 289 it states that the expression of 13 genes is markedly altered, and in line 290 it further states the same for C-Myc, Rex1, Klf4, Mef2, Ils1, Sox17, and Nestin. Do you singled out these genes here because I wanted to emphasize the contents of line 291? If so, it is easier to understand how to write line 293.

: The sentence that ‘C-Myc, Rex1, Klf4, Mef2, Ils1, Sox17, and Nestin showed a significant change’ was deleted to reduce reader fatigue.

 Comment 8.

Line297:None of the changes by the SA treatment were restored by DPA except for the c-Myc and Mef2 gene (Fig 5 B and E).ï¼›I think it is better not to use the word “restored” since this description also does not mean that what was changed has been restored (since it is a simultaneous exposure).

: The term ‘restored’ has been replaced to ‘protected’.

 Comment 9.

Line366-368:In Figure 2A, the brachyury gene, which is expressed at the mesoderm state, tends to increase and then decrease gradually. The present results support previous reports that the brachyury gene is expressed mostly during the early stages of differentiation.; This explanation helped us to understand why the expression in the control group is reduced.

It seems that this is the only brachyury gene in Fig. 2 that is affected by arsenic in the initial stage (4Day). Please discuss this phenomenon further.

: We further discussed this part by comparing the expression of other genes.

 Comment 10.

Line366-367:As already commented, the use of "recovered" or "restored" is not appropriate in the case of simultaneous exposure. Please revise the wording to make it less misleading, e.g., simultaneous exposure suppressed the effects of SA.

: We have replaced the term to ‘preserved’.

Thank you for your kind and detailed review. This allowed us to fix what we missed. Replaced the term with an example. Removed duplicate sentence referring to 'recover or restore' to 'protect or defend' and updated reference. Also, everything else has been corrected and accounted for. Then we look forward to your positive decision.

Round 2

Reviewer 1 Report

The authors have revised most of the comments. However, there are still a few questions.

Referring to my previous comment no 6. “Line 233-236. The authors should clarify which particular subunit of each ETC complex they measured. And to clarify where they are encoded, in nuclear DNA or mtDNA.” 

It would be more correct to clarify that expression of Ndufs1 gene was measured to assess the level of complex I, Sdha for complex II, etc.

“The five complexes were measured by real-time PCR at the mitochondrial DNA”.

All subunits under study cannot be encoded in mtDNA, since mtDNA does not contain the genes that encode complex II. BLAST analysis showed that all the subunits under investigation are encoded by nuclear DNA. Please clarify.

Author Response

Jan 18th 2022

Response to Reviewer 1:

We have made the following editorial changes in response to his/her insightful comments.

We tried to revise the manuscript following your comments.

Review Comments to the Author

Comment 1.

The authors have revised most of the comments. However, there are still a few questions.

Referring to my previous comment no 6. “Line 233-236. The authors should clarify which particular subunit of each ETC complex they measured. And to clarify where they are encoded, in nuclear DNA or mtDNA.”

It would be more correct to clarify that expression of Ndufs1 gene was measured to assess the level of complex I, Sdha for complex II, etc.

“The five complexes were measured by real-time PCR at the mitochondrial DNA”.

All subunits under study cannot be encoded in mtDNA, since mtDNA does not contain the genes that encode complex II. BLAST analysis showed that all the subunits under investigation are encoded by nuclear DNA. Please clarify.

: I am totally agree with the reviewer’s comment. I think all the three questions are on the same page. According to the reviewer’s comment, we revised the phase in the context.

“The changes of mRNA expression of five complexes due to SA treatment were measured by real-time PCR at the resting EB state after SA treatment for four days. The mitochondrial mRNA expression level of NADH:Ubiquinone Oxidoreductase Core Subunit S1 (Ndufs1) was measured to assess the level of Complex I, Ubiquinol-Cytochrome C Reductase, Rieske iron-sulfur polypeptide 1 (Uqcrfs1) for Complex III, Cytochrome c oxidase subunit 5A (Cox5a) for Complex IV, and ATP synthase F1 subunit delta (Atp5d) for Complex V. The Complex II, which are encoded nuclear mRNA, was assessed by measuring nuclear mRNA expression of succinate dehydrogenase complex flavoprotein subunit A (Sdha).”

*Thanks for your valuable comment and makes our manuscript better.
